# Unifying Model-Based and Model-Free Reinforcement Learning with Equivalent Policy Sets

## Abstract

Model-based and model-free reinforcement learning (RL) each possess relative strengths that prevent either algorithm from strictly dominating the other. Model-based RL often offers greater data efficiency, as it can use models to evaluate many possible behaviors before choosing one to enact. However, because models cannot perfectly represent complex environments, agents that rely too heavily on models may suffer from poor asymptotic performance. Model-free RL avoids this problem at the expense of data efficiency. In this work, we seek a unified approach to RL that combines the strengths of both algorithms. To this end, we propose *equivalent policy sets* (EPS), a novel tool for quantifying the limitations of models for the purposes of decision making. Based on this concept, we propose *Unified RL*, a novel RL algorithm that uses models to constrain model-free RL to the set of policies that are not provably suboptimal, according to model-based bounds on policy performance. We demonstrate across a range of benchmarks that Unified RL effectively combines the relative strengths of both model-based and model-free RL, in that it achieves comparable data efficiency to model-based RL and exceeds the data efficiency of model-free RL, while achieving asymptotic performance similar or superior to that of model-free RL. Additionally, we show that Unified RL outperforms a number of existing state-of-the-art model-based and model-free RL algorithms, and can learn effective policies in situations where either model-free or model-based RL alone fail.

## 1 Introduction

Recent successes in model-based reinforcement learning (MBRL) have demonstrated the enormous value that learned representations of environmental dynamics (*i.e.*, models) can confer to autonomous decision making. For example, models allow agents to evaluate many possible future behaviors, without requiring additional expensive and potentially dangerous environmental interactions. This process is referred to as *planning*, and is a cornerstone of autonomous decision making. Models also hold the potential to facilitate cross-task knowledge transfer (Killian *et al.*, 2017) and intelligent exploration (Lowrey *et al.*, 2018; Sekar *et al.*, 2020; Mehta *et al.*, 2021; 2022). In practice, MBRL algorithms often achieve higher data efficiency than their model-free counterparts (Deisenroth & Rasmussen, 2011; Heess *et al.*, 2015; Gal *et al.*, 2016a; Chua *et al.*, 2018; Janner *et al.*, 2019; Hafner *et al.*, 2019; 2020; Lin *et al.*, 2023).

Although useful, models come with their own set of drawbacks. Because models typically posses limited representational capacity, they will always fall short of capturing the full complexity of the real environmental dynamics, which may help explain why MBRL often fails to match the asymptotic performance of model-free RL (MFRL) (Wang *et al.*, 2019). This limitation of models is exacerbated by the *objective mismatch problem* (Lambert *et al.*, 2020): model-learning objectives typically used in MBRL, which are based on some generic measure of accuracy, are often misaligned with the overall goal of increasing reward, which has been shown to negatively impact MBRL performance in practice (Agarwal *et al.*, 2021).

Several recent approaches have attempted to address objective mismatch by deriving model-learning objectives that are more aligned with the overall RL objective, to enable learned models to be more

useful for policy improvement (Joseph *et al.*, 2013; Luo *et al.*, 2018; Lambert *et al.*, 2020; Rajeswaran *et al.*, 2020; Chow *et al.*, 2020; Grimm *et al.*, 2020; D'Oro *et al.*, 2020; Eysenbach *et al.*, 2022; Ghugare *et al.*, 2022). However, because practical models will always differ from the true dynamics by some degree, we hypothesize that over-reliance on models will invariably result in some degree of suboptimality. For this reason, we take an alternative approach to addressing the objective mismatch problem. We seek to develop agents that understand the limitations of their models, allowing them to switch to an alternative (*e.g.*, a model-free) learning paradigm in situations where models are not useful for policy improvement. We hypothesize that such an agent would enjoy the benefits of both model-based and model-free learning. To this end, we propose *equivalent policy sets* (EPS), a novel tool for quantifying the limitations of a model for estimating optimal behaviors. We define the EPS as the set of all policies that are not *provably suboptimal*, using bounds on the performance of candidate policies, computed using the model. Intuitively, the EPS captures the usefulness of a particular model class for discerning optimal from suboptimal policies.

Based on the concept of the EPS, we propose *Unified RL*, a principled approach to combining MBRL and MFRL that takes advantage of their relative strengths. Unified RL constrains the policy found by MFRL (*e.g.*, soft actor-critic) to lie within the set of non-provably suboptimal policies (the EPS). Here, models are used as a sort of "pre-filtering" step that eliminates provably suboptimal policies from consideration by MFRL. Unified RL leverages the ability of models to rapidly rule-out suboptimal candidate behaviors, while avoiding limitations on asymptotic performance that they introduce.

We show empirically that Unified RL is able to combine the benefits of both model-based and model-free RL on a range of challenging continuous control benchmarks. Furthermore, we show that Unified RL outperforms a wide range of state-of-the-art model-based and model-free RL algorithms. Finally we show that Unified RL is robust to failure of either its model-based or model-free components. Specifically, when distractors are introduced that prevent the agent from learning well-aligned models, Unified RL continues to make learning progress using model-free policy updates. On the other hand, when poorly selected model-free hyperparameters are used that cause MFRL to fail, Unified RL resorts to MBRL.

## 2 BACKGROUND

We represent the environment with which the agent interacts as a Markov decision process (MDP) with initial state distribution $s_0 \sim p_0(s_0)$, state transition dynamics $s_{t+1} \sim T(s_{t+1}|s_t, a_t)$, reward function $r_t \sim R(r_t|s_t, a_t)$ for $t \in \{0, ..., T\}$, and discount factor $\gamma \in [0, 1]$. For simplicity, we assume $\gamma = 1$ and hence ignore it in future exposition. We consider continuous control problems, wherein the agent learns a policy $\pi \in \Pi$ where $\pi : \mathcal{S} \times \mathcal{A} \to [0, \infty)$ is a state-dependent probability density function over a real-valued action space.

In this work, we formulate the RL problem in Bayesian terms, although the approach is not restricted to using Bayesian algorithms. We are therefore concerned with the Bayesian posterior over state transition and reward functions, given by $p(w|D) = {}^{p(D|w)p(w)}/{}_{p(D)}$, where $D$ is comprised of data observed thus far in the environment, $w$ denotes a parameter vector that parameterizes both the state transition and reward functions, and $p(w)$ is our prior. The prior represents our belief about the dynamics before observing data $D$, and can be informed by domain-specific knowledge or from previous tasks. In this work we do not assume that we possess any prior knowledge, and therefore choose a generic prior (Sec. 3.2). We denote our models of the state transition function and reward function, conditioned on a certain parameter vector $w$, as $p(s'|s, a, w)$ and $p(r|s, a, w)$, respectively. The distribution of trajectories $\tau$ given a particular policy $\pi$ and parameters $w$ is given by $p(\tau|\pi, w) = p(s_0)\pi(a_0|s_0)p(r_0|s_0, a_0, w)\prod_{t=1}^{T} p(s_t|s_{t-1}, a_{t-1}, w)\pi(a_t|s_t)p(r_t|s_t, a_t, w)$. Our inferred posterior distribution over trajectories given the available data $D$ and a policy $\pi$ is given by $p(\tau|D) = \mathbb{E}_{p(w|D)}p(\tau|\pi, w)$. We denote the expected return of $\pi$ given a particular parameter vector $w$ as $J(\pi|w) = \mathbb{E}_{p(\tau|\pi, w)} \left[ \sum_{t=0}^{T} r_t \middle| \pi, w \right]$. Finally, we define the *Bayesian return* of a policy $\pi$ to be the expected sum of rewards achieved by $\pi$, in expectation over our Bayesian posterior over trajectories

$$J(\pi|D) = \mathbb{E}_{p(\tau|D,\pi)} \left[ \sum_{t=0}^{T} r_t \middle| \pi, D \right]. \tag{1}$$

This is the quantity that our approach to Bayesian RL attempts to maximize. We refer to a policy that maximizes the Bayesian return $\pi^* \in \arg\max_{\pi \in \Pi} J(\pi|D)$ as the *Bayes-optimal policy*. Similarly, we refer to any policy $\pi \notin \arg\max_{\pi \in \Pi} J(\pi|D)$ as *Bayes-suboptimal*.

For many interesting model classes, exact Bayesian posteriors are intractable, and must therefore be approximated with some tractable distribution family. We denote approximate posteriors with $q(w; \theta) \in \mathcal{Q}$, where $\theta$ denotes the parameters of the distribution. For example, if $q$ is a multivariate normal distribution, $\theta$ may contain the mean vector and variance matrix. We henceforth refer to $q$ as our *model*, because it encodes our learned representation of (our posterior over) the environmental dynamics.

## 3 UNIFYING MODEL-BASED AND MODEL-FREE REINFORCEMENT LEARNING

Here we introduce the notion of *equivalent policy sets* (EPS) as a tool for quantifying the limitations of models for the purposes of approximating optimal policies. Subsequently, we describe *Unified Reinforcement Learning*, which builds on the concept of the EPS to combine the strengths of model-based and model-free RL.

### 3.1 EQUIVALENT POLICY SETS

To achieve our ultimate goal of developing agents that can flexibly switch between model-free and model-based learning, agents must understand the limitations of models for evaluating and improving policies. To this end, we propose *equivalent policy sets* (EPS) as a tool for quantifying the usefulness of a model for discerning optimal from suboptimal policies. More precisely, we define the EPS $\Pi_E(\theta, D) \subseteq \Pi$ to be the set of policies that are not provably Bayes-suboptimal, using a model with parameters $\theta$ and available data $D$. To prove the suboptimality of a particular policy $\pi$, we use our model to compute a lower bound on (a function $f$ of) the improvement in Bayesian return of a *new* policy $\pi'$ over $\pi$,

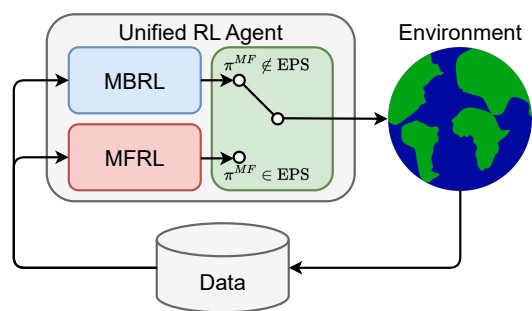

Figure 1: Unified RL combines model-based and model-free RL using the equivalent policy set (EPS). At each iteration, data from a shared buffer are used to update a model-based policy and a model-free policy. We then check whether the model-free policy is contained within the EPS, that is, the set of policies that cannot be proven to be suboptimal, according to bounds on policy performance computed using the model. If the model-free policy is within the EPS, it is used to collect another episode of data in the environment, which is added to the data buffer. Otherwise, the model-based policy is used to collect more data.

$$\mathcal{L}(\pi, \pi', \theta, D) \le f((J(\pi'|D) - J(\pi|D))), \qquad (2)$$

where $f$ is a monotonically increasing function. Although one could use any such $\mathcal{L}$, in this work we take $\mathcal{L}$ to be of the form

$$\mathcal{L}(\pi, \pi', \theta, D) = \mathbb{E}\left[ f\left( \frac{p(D|w)p(w)}{q(w; \theta)} \left( J(\pi'|w) - J(\pi|w) \right) \right) \right], \qquad (3)$$

which we derive in the Sec. A.1 of the Appendix using Jensen's inequality. This particular form of $\mathcal{L}$ requires $f$ to be concave, and is closely related to $f$-divergences, a generalization of the widely used KL and Réyni divergences (Li & Turner, 2016; Wan *et al.*, 2020). In the closely-related field of variational inference, the effect of the choice of $f$ is an active area of research, and gives rise to various divergence metrics Kingma & Welling (2013); Burda *et al.* (2015); Li & Turner (2016); Dieng *et al.* (2017); Chen *et al.* (2018); Wan *et al.* (2020). In this work we primarily consider $f = \log$, as this is the most well-studied choice of $f$ Blei *et al.* (2017). $\mathcal{L}$ is tight (*i.e.*, inequality 2 holds with equality) when $q(w; \theta) \propto p(D|w)p(w)(J(\pi'|w) - J(\pi|w))$. Note that, although $\mathcal{L}$

depends on the parameters $\theta$ of the *approximate* posterior $q$, inequality 2 bounds the *exact* difference in Bayesian return between $\pi'$ and $\pi$.

Inequality 2 allows us to prove the suboptimality of any policy $\pi$ for which there exists a new policy $\pi'$ (in the same domain as $\pi$) such that $\mathcal{L}(\pi, \pi', \theta, D) > f(0)$, because this condition implies that $\pi'$ achieves higher Bayesian return than $\pi$, and therefore $\pi$ is not Bayes-optimal. We can therefore use $\mathcal{L}$ to construct the EPS, which we define to be *the set of all policies $\pi$ for which there does not exist a provably better $\pi' \in \Pi$*, using model parameters $\theta$ and data $D$,

$$\Pi_E(\theta, D) = \{\pi : \max_{\pi' \in \Pi} \mathcal{L}(\pi, \pi', \theta, D) \leq f(0)\}. \tag{4}$$

**Equivalent Policy Sets for Understanding the Limitations of Models** In the limit of an infinitely expressive model (that is, $q$ can represent any posterior over $w$), $\mathcal{L}$ is tight, meaning that the EPS reduces to a singleton set that contains only the Bayes-optimal policy. However, limitations in modeling resources make this practically infeasible, and in general the model will always contain some inaccuracies. Existing approaches to MBRL largely have not dealt with this problem, and instead treat the model's approximation of the optimal policy as ground-truth. This can result in highly suboptimal policies, especially when the model is misaligned (Lambert *et al.*, 2020; Agarwal *et al.*, 2021). The EPS addresses this problem by quantifying how inaccuracies in our imperfect model translate into uncertainty about the optimal policy, where this uncertainty is represented as a *set* of policies that *may* be optimal, according to our model. Limitations in model class prevent $q$ from matching the ideal posterior, causing $\mathcal{L}$ to be loose and thereby increasing the size of the EPS. By maintaining this set, we avoid over-reliance on the model, and open the possibility of using an alternative learning paradigm such as MFRL to choose a policy to deploy. This intuition provides the basis for Unified RL, which we describe in the next section.

## 3.2 Unified Reinforcement Learning

Unified RL builds on the concept of the EPS introduced in the previous section, and is summarized in Alg. 1 and Fig. 1. Unified RL can be thought of as a model-free RL algorithm, where the policy is constrained to lie within the EPS. Through this constraint, Unified RL is able to eliminate many provably suboptimal policies from consideration, thus retaining the data-efficiency benefits of MBRL. However, because Unified RL uses the model only to identify the set of policies that *may* be optimal rather than to estimate a single optimal policy, it avoids over-reliance on the model, and thus avoids the objective mismatch problem associated with typical MBRL approaches. Constraining the model-free policy to lie within the EPS does not in principle prevent MFRL from discovering the Bayes-optimal policy, as the Bayes-optimal policy will always lie within the EPS regardless of the model used to compute the EPS.

---

**Algorithm 1** Unified RL

1: **Given:** initial dataset $D$
2: **for** each iteration **do**
3: $\quad \pi^{MB}, \theta =$ MBRL$(D)$
4: $\quad \pi^{MF} =$ SAC$(D)$
5: $\quad$ Estimate $\hat{\mathcal{L}}(\pi^{MF}, \pi^{MB}, \theta, D)$
6: $\quad$ **if** $\hat{\mathcal{L}} > -\infty$ **then**
7: $\quad\quad \pi = \pi^{MB}$
8: $\quad$ **else**
9: $\quad\quad \pi = \pi^{MF}$
10: $\quad$ **end if**
11: $\quad$ **for** time step t=0,...,T **do**
12: $\quad\quad a_t \sim \pi(a_t|s_t)$
13: $\quad\quad s_{t+1}, r_t =$ env.step$(a_t)$
14: $\quad\quad D \leftarrow D \cup \{s_t, a_t, r_t, s_{t+1}\}$
15: $\quad$ **end for**
16: **end for**

---

We take a simple approach to combining model-based and model-free RL using the EPS, and leave more complex variants to future work. Before each episode, an MBRL and an off-policy MFRL algorithm use the available data $D$ to compute what we refer to as the *model-based policy* $\pi^{MB}$ and the *model-free policy* $\pi^{MF}$, respectively. Subsequently, the agent checks whether the model-free policy is within the EPS; that is, it checks whether or not a lower bound can be constructed using the model that proves that the model-based policy achieves higher Bayesian return than the model-free policy. If the model-free policy is within the EPS, the agent executes it in the real environment to collect one episode of new data. If not, the agent instead executes the model-based policy, which is guaranteed to be within the EPS. The new data are then added to the shared data buffer, and the

entire process repeats. Note that this approach does not require the EPS to be represented explicitly. Instead, the EPS is maintained *implicitly* in the sense that the lower bound in equation 2 provides a condition that allows one to check whether a given policy is within the EPS. We describe the individual components of our approach in more detail below, with additional details in Sec. A.2 of the Appendix.

**Model-Based RL**  The MBRL component of our algorithm proceeds in two distinct steps: model training and policy training. During the model training step, we estimate the posterior parameters $\theta$ by fitting a Bayesian LSTM dynamics model to our environmental data $D$, by maximizing an evidence lower bound on data log likelihood (Kingma *et al.*, 2015; Gal & Ghahramani, 2016b;a),

$$\mathcal{L}_{\text{model}}(\theta, D) = \mathbb{E}_{w \sim q(w;\theta)} \left[ \sum_{i=1}^{|D|} \sum_{t=1}^{T} \log p(s_{t+1}^{(i)}, r_t^{(i)} | s_{\leq t}^{(i)}, a_{\leq t}^{(i)}, w) \right] - D_{KL}\big(q(w;\theta)||p(w)\big). \quad (5)$$

Specifically, we use the *binary dropout* formulation of Bayesian LSTMs (Gal & Ghahramani, 2016a), wherein sampling a weight from the posterior $w \sim q(w;\theta)$ is accomplished by sampling a binary dropout mask from a fixed Bernoulli distribution (Gal & Ghahramani, 2016b). In this formulation, the prior $p(w)$ is approximately a Normal distribution, while the posterior is a Bernoulli Gal *et al.* (2016b). Our dynamics model $p(s_{t+1}^{(i)}, r_t^{(i)} | s_{\leq t}^{(i)}, a_{\leq t}, w)$ is a Gaussian distribution over next state $s_{t+1}$ and reward $r_t$ with a diagonal covariance matrix, given the states $s_{\leq t}$ and actions $a_{\leq t}$ at all previous timesteps. The choice to represent state transition dynamics as Gaussians with diagonal covariance matrices is similar to past work (Gal *et al.*, 2016a; Chua *et al.*, 2018; Gamboa Higuera *et al.*, 2018; Chow *et al.*, 2020; Eysenbach *et al.*, 2022; Freed *et al.*, 2023), with the primary difference being that our dynamics model is recurrent. Specifically, we use an LSMT dynamics model, as we found this to yield more stable gradient-based policy optimization compared to a simple feed-forward MLP.

During the policy training step, we train a Tanh-Gaussian policy (Haarnoja *et al.*, 2018) to maximize the expected cumulative reward predicted by our model. Depending on the environment, we found that one of two methods yielded the best results. In both methods, we start by sampling a set of weights from our approximate posterior (which corresponds to sampling a set of dropout masks). In the first method, for each weight, we sample a set of initial states from the initial state distribution, which we assume to be known. Subsequently, we sample a full $T$-length trajectory, starting from each initial state, by iteratively sampling actions from the policy, followed by a reward and state transition from the model. Given a batch of sampled trajectories, we compute the policy loss as the negative total reward along the trajectory averaged across sampled trajectories, plus a policy entropy bonus. Similar to Gamboa Higuera *et al.* (2018), we found that gradient clipping stabilized policy optimization and improved results. We refer to this method as full-trajectory policy training, because full-length trajectories are rolled out.

The second method of policy training that we employ is identical to that used by Hafner *et al.* (2019), with the slight modification that trajectories are sampled using various dropout masks, and trajectories are sampled in raw state space as opposed to latent space. In summary, states are sampled uniformly from the data buffer, and trajectory segments of length $H = 16$ are sampled starting from those states. Value estimates are then computed using a critic network and the predicted trajectory rewards. The critic is then updated to produce more accurate value estimates, and the policy is updated to produce higher value estimates. In either case, the dropout mask that we use to sample a particular trajectory is held constant during the entire trajectory; this is to reflect the fact that even though there is uncertainty in the dynamics model parameters $w$, the parameters do not change during a single trajectory (Gal & Ghahramani, 2016a).

**Model-Free RL**  We use Soft Actor-Critic (Haarnoja *et al.*, 2018) as the off-policy MFRL component of our algorithm. We found that standard SAC performed poorly when run off-policy; therefore, we incorporate two modifications suggested by Ball *et al.* (2023) that we found yielded superior off-policy performance while preserving SAC's on-policy performance. Specifically, we used layer normalization in our Q networks, and omit the entropy term from the Q network loss.

**Lower Bound Estimation**   Using the posterior parameters $\theta$ obtained during the model learning process and $f = \log$, it is possible to compute a Monte-Carlo estimate of $\mathcal{L}$ as

$$\hat{\mathcal{L}}(\pi^{MF}, \pi^{MB}, \theta, D) = \sum_{i=1}^{K} \Big( \log \frac{p(D|w_i)p(w_i)}{q(w_i; \theta)} + \log \Big( \hat{J}(\pi^{MB}|w_i) - \hat{J}(\pi^{MF}|w_i) \Big) \Big), \quad (6)$$

for $w_1, ..., w_K \sim q(w; \theta)$. Here, $p(D|w_i)$ is the probability of all state transitions and rewards in the dataset given parameters $w_i$, and $\hat{J}(\pi^{MB}|w)_i$ and $\hat{J}(\pi^{MF}|w_i)$ are themselves Monte-Carlo estimates of the expected return for the model-based and model-free policies respectively, computed by rolling out a batch of $M$ trajectories from the model using parameters $w_i$ and policies $\pi^{MB}$ and $\pi^{MF}$, respectively. More details on the estimation of this bound are provided in Sec. A.2 of the Appendix.

We review some useful properties of $\hat{\mathcal{L}}$. As $K \to \infty$ and $M \to \infty$, by the law of large numbers, $\hat{\mathcal{L}} \to \mathcal{L}$. However, for $K \to \infty$ and finite $M$, by Jensen's inequality, $\hat{\mathcal{L}} \leq \mathcal{L}$. This property does not change our algorithm in principle, because $\hat{\mathcal{L}}$ for finite $M$ is still a lower bound on $\log(J(\pi^{MB}|D) - J(\pi^{MF}|D))$. The only practical implication in using $\hat{\mathcal{L}}$ in place of $\mathcal{L}$ is that the algorithm becomes more conservative, preferring model-free RL more often, as it becomes more difficult to prove that the model-based policy achieves a higher Bayesian return. When $K$ is also finite, $\hat{\mathcal{L}}$ is stochastic, and we can no longer say it is strictly a lower bound on $\log(J(\pi^{MB}|D) - J(\pi^{MF}|D))$, though on average it is. Practically, the stochasticity of $\hat{\mathcal{L}}$ injects some randomness into policy selection. We did not find this to be an issue as long as a large enough value of $K$ and $M$ were used.

To check if the model-free policy is in the EPS, we must check whether $\hat{\mathcal{L}}(\pi^{MB}, \pi^{MF}, \theta, D) > \log(0) = -\infty$. Note that in equation 6, all terms except $\log \Big( \hat{J}(\pi^{MB}|w_i) - \hat{J}(\pi^{MF}|w_i) \Big) \Big)$ will be defined and finite. However, as $\hat{J}(\pi^{MB}|w_i) - \hat{J}(\pi^{MF}|w_i) \to 0$ from the right, $\log \big( J(\pi^{MB}|w_i) - J(\pi^{MF}|w_i) \big) \to -\infty$. Therefore, this term dominates $\hat{\mathcal{L}}$ when the model-free policy is on or near the boundary of the relevant set, allowing us to simply check whether $\hat{J}(\pi^{MB}|w_i) - \hat{J}(\pi^{MF}|w_i) > 0, \quad \forall i = 1, ..., K$. This property is particularly convenient because it allows us to ignore the $\log p(D|w_i)$ term, which would normally require calling the model on the entire dataset.

## 4   Experiments

Our experiments seek to answer three questions:

1. Can Unified RL successfully combine the strengths of model-based and model-free RL?
2. Does Unified RL perform favorably compared to state-of-the-art prior work?
3. Is Unified RL effective in situations where either MBRL or MFRL alone fail?

We address questions 1 and 2 in Sec. 4.1, and question 3 in Sec. 4.2 and Sec. 4.3. In our experiments, we consider a range of challenging continuous control tasks from the OpenAI gym benchmark suite Brockman *et al.* (2016), Deepmind Control Suite (DMC) Tassa *et al.* (2018), and the ROBEL robotics benchmark suite Ahn *et al.* (2020). Specifically, we consider OpenAI gym Hopper, Walker, Ant, and Half-Cheetah, as well as DMC Cartpole Swingup and ROBEL DClawTurnFixed. We make two modifications to the standard environments for the sake of simplicity. First, we disabled early episode termination in the OpenAI gym tasks, as early termination has been shown to cause issues for MBRL (Wang *et al.*, 2019). Second, we focus on short time horizon tasks; specifically, we consider episodes of length $T = 100$ for all OpenAI gym and DMC tasks, except for Hopper and Cartpole, which we considered episodes of length $T = 200$. We found that these episodes were sufficiently long to allow agents to learn the desired behaviors.

### 4.1   Data Efficiency and Asymptotic Performance

To empirically evaluate the effectiveness of Unified RL at combining the strengths of model-based and model- free RL, we compare Unified RL to its constituent model-based and model-free compo-

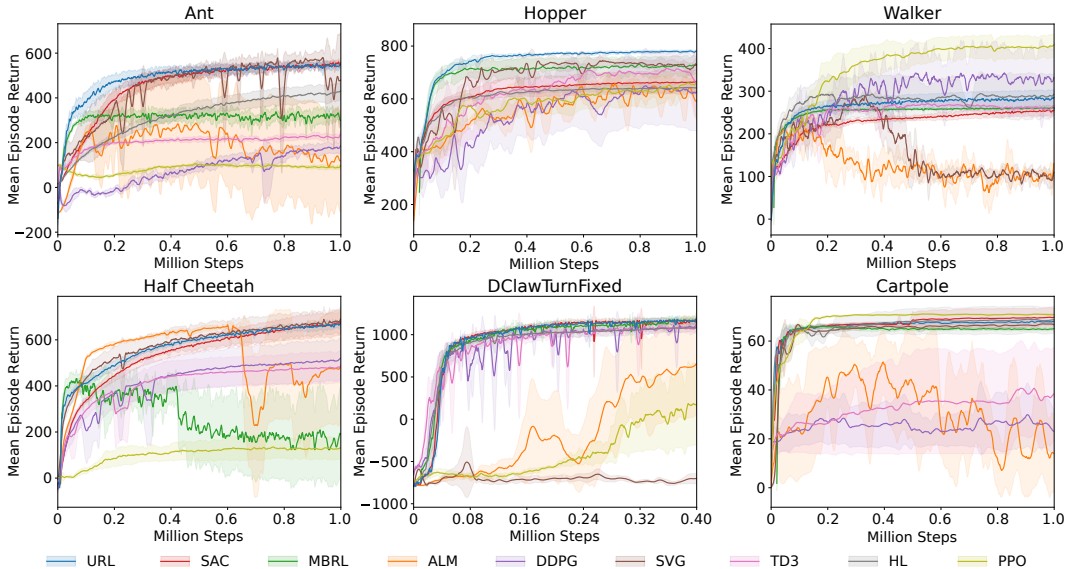

Figure 2: Training curves on benchmark tasks. Solid lines indicate the average return per episode across 5 runs, while shaded regions denote 95% confidence intervals. We find that Unified RL successfully combines the strengths of both model-based and model-free RL. In environments where either MBRL or SAC strictly dominates the other, Unified RL at least matches the better of these two algorithms. In situations where MBRL learns faster initially but is eventually surpassed by SAC, Unified RL achieves higher performance than either algorithm alone. Additionally, Unified RL also performs favorably compared to the other baselines, and is the only algorithm we tested that consistently performs well across all tasks.

Table 1: Mean episode return on benchmark tasks. Reported below are episodes returns averaged across the entire training process for 5 distinct random seeds, with 95% confidence intervals. The final column is the average rank that each algorithm achieves across all environments. We find that Unified RL often achieves higher mean episode return compared to either MBRL or SAC, indicating that Unified RL is able to combine the strengths of both algorithms. Additionally, Unified RL is the only algorithm we tested that performed consistently well across all tasks, being the most high-ranking algorithm on average.

| | Ant | Hopper | Walker | Half Cheetah | Cartpole | DClawTurnFixed | Avg Rank |
|---|---|---|---|---|---|---|---|
| Unified RL | **493 ± 9.9** | **750 ± 2.4** | 267 ± 5.9 | **571 ± 3.5** | 66.7 ± 0.3 | 963 ± 11 | **2.33** |
| SAC | 457 ± 7.9 | 633 ± 14 | 229 ± 2.1 | 540 ± 12 | 67.5 ± 0.9 | 970 ± 11 | 3.5 |
| MBRL | 311 ± 10 | 705 ± 16 | 253.2 ± 0.8 | 263 ± 48 | 64.6 ± 0.2 | **989 ± 16** | 4.5 |
| ALM | 188 ± 77 | 563 ± 6.2 | 127 ± 7.6 | 520 ± 27 | 31.5 ± 2.9 | −177 ± 111 | 7.33 |
| DDPG | 76 ± 6.4 | 542 ± 42 | 293 ± 21 | 422 ± 34 | 25.3 ± 3.7 | 877 ± 17 | 6.83 |
| SVG | 443 ± 15 | 684 ± 14 | 163 ± 6.6 | **582 ± 21** | 65.0 ± 0.3 | −705 ± 14 | 4.83 |
| TD3 | 204 ± 3.8 | 632 ± 27 | 250 ± 4.2 | 412 ± 23 | 32.6 ± 7.8 | 902 ± 26 | 5.83 |
| HL | 319 ± 12 | 616 ± 4.5 | 280 ± 19 | **569 ± 8.2** | 66.6 ± 1.8 | 889 ± 29 | 4.16 |
| PPO | 85 ± 1.6 | 582 ± 18 | **356 ± 12** | 104 ± 16 | **69.2 ± 0.1** | −404 ± 56 | 5.16 |

nents alone. We also compare to several prior state-of-the-art approaches: Aligned Latent Models (ALM) (Ghugare *et al.*, 2022), Deep Deterministic Policy Gradient (DDPG) (Lillicrap *et al.*, 2015), Twin Delayed DDPG (TD3) (Fujimoto *et al.*, 2018), Proximal Policy Optimization (PPO) (Schulman *et al.*, 2017), Stochastic Value Gradient (SVG) (Heess *et al.*, 2015), and Hybrid Learning (HL) (Pinosky *et al.*, 2023). DDPG, TD3, and PPO are state-of-the-art model-free methods. ALM, SVG and HL are high-performing algorithms that combine aspects of model-based and model-free RL.

We report our results in two ways. First, we show mean episode return vs. number of environmental steps in Fig. 2. Here, solid lines indicate average episode return averaged across 5 independent random seeds, while shaded regions denote 95% confidence intervals. Second, we report the average episode return across the entire training process for each algorithm, which is equivalent to area under the learning curve normalized by number of training episodes, in Table **??**. This statistic is relevant because it blends both data efficiency and asymptotic performance into a single scalar

performance metric. Here again we report the average return across 5 random seeds, with 95% confidence intervals.

Our first observation is that Unified RL succeeds at combining the strengths of its two constituent algorithms. In cases where one algorithm strictly dominates, such as Hopper and Walker, we see that Unified RL does at least as well as the better-performing constituent. Moreover, we find that in environments such as Ant and Half-Cheetah, where MBRL learns rapidly initially but is eventually surpassed by SAC, Unified RL achieves higher performance than either algorithm alone. This finding indicates that Unified RL enables a synergy between MBRL and MFRL that is superior to simply running both algorithms separately and picking the best one at each timestep. We additionally observe that of all the algorithms we tested, Unified RL was unique in that it performed well across all tasks. Interestingly, ALM seemed to suffer from instability, possibly due to issues in Q learning caused by the shorter episode lengths that we use in our experiments.

## 4.2 ROBUSTNESS TO MODEL MISALIGNMENT

One of our central claims is that Unified RL helps avoid the objective mismatch problem by allowing the agent to switch to MFRL when the model is misaligned (that is, ill-suited to helping the agent improve its policy). To test this claim, we evaluate Unified RL on a task that we designed to induce model misalignment in MBRL. Recall that distractors are components of the observation that are predictable but task-irrelevant. Distractors exacerbate model misalignment, because typical model-learning objectives do not prioritize the modeling of task-relevant observation components over the task-irrelevant distractors. This results in models that do not accurately represent the task-relevant components. In our experiments, we appended

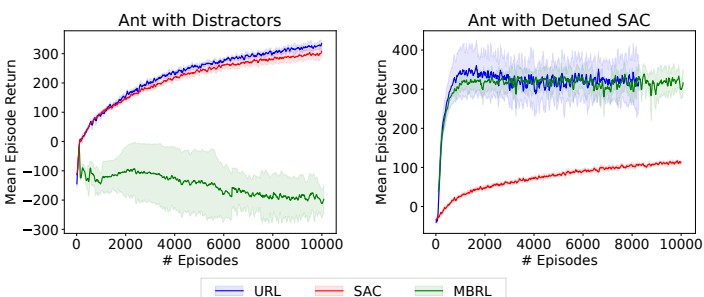

Figure 3: Unified RL is robust to both failures in model-based and model-free RL. (Left) We induce model misalignment by introducing distractors into the observations of the Ant environment, which prevents model-based RL from learning effectively. We find that, in this situation, Unified RL matches the performance of model-free RL, indicating that Unified RL is robust to failure of model-based RL. (Right) We induce failure of model-free RL (SAC) by increasing the entropy bonus to a suboptimal value. In this situation, we find that Unified RL matches the performance of model-based RL, indicating that Unified RL is robust to failure of model-free RL.

time-dependent sinusoids of fixed frequency to the observations. Sinusoids were grouped together into groups of 10, where all 10 sinusoids in a group had the same phase. Each group was assigned a random phase, preventing the model from simply memorizing the distractors. Five such groups were appended to the observations. The hyperparameters used for SAC, MBRL, and Unified RL for this experiment were identical to those used in the original Ant environment.

The reward curves for this experiment are shown in Fig. 3. We observe that MBRL utterly fails to make learning progress in the presence of distractors, while MFRL is relatively unphased. Unified RL performs slightly better than MFRL, indicating that it is able to effectively fall back on MFRL when its model is misaligned.

## 4.3 ROBUSTNESS TO FAILURES OF MODEL-FREE RL

We do not expect MFRL to always achieve higher asymptotic performance than MBRL in all environments; for example, MFRL may fail to escape a poor local minimum or have poorly tuned hyperparameters. Unified RL has the advantage over other approaches such as MBRL with Model-Free Fine Tuning Nagabandi *et al.* (2018), which runs MBRL for a manually specified number of episodes before switching to MFRL, in that Unified RL only switches to MFRL when the model-based policy isn't provably superior. Therefore, in situations where MFRL fails to learn effectively, we expect Unified RL to utilize model-based learning exclusively. To test this claim, we compare the performance of Unified RL to MBRL and SAC in the Ant environment, where the entropy penalty for both SAC and the SAC component of Unified RL was set far higher than its ideal value. As expected, this prevented SAC from learning effectively, both alone and within Unified RL. Indeed we found that Unified RL recognized that SAC was ineffective at solving the task, instead relied exclusively on MBRL.

## 5 RELATED WORK

Similar to Duff (2002); Deisenroth & Rasmussen (2011); Gal *et al.* (2016a); Chua *et al.* (2018); Gamboa Higuera *et al.* (2018); Mehta *et al.* (2021; 2022), we consider a Bayesian formulation of MBRL. The characteristic feature of these approaches is an explicit representation of uncertainty in their estimate of the environmental dynamics. Gal *et al.* (2016a), Depeweg *et al.* (2017), and Gam-

boa Higuera *et al.* (2018) are most similar to our approach, in that they use Bayesian neural networks (BNNs) to represent beliefs over dynamics, and learn policies by backpropagating gradients through model rollouts.

Several recent approaches have been proposed for combining model-based and model-free RL. For example, Hybrid Learning (Pinosky *et al.*, 2023) used a learned dynamics model to determine an optimal time to switch between a planned action sequence and a policy learned using MFRL. Stochastic Value Gradients (Heess *et al.*, 2015) proposed a spectrum of policy gradient algorithms that range from model-free methods with value functions to model-based methods without value functions. Finally, Model-Based RL with Model-Free Fine Tuning initialized MFRL with a policy trained for a fixed number of episodes using MBRL. The primary drawback to these approaches that is addressed in our work is that they use hard-coded or heuristic methods for selecting which learning modality to use in a given situation, rather than switching based on a measure of the model's ability to contribute to policy improvement.

Recent approaches for improving model alignment in MBRL optimized policies with respect to lower bounds similar to $\mathcal{L}$. For example, Luo *et al.* (2018) considered iteratively constructing lower bounds that hold locally in policy space, which is then optimized jointly with respect to both the model and policy. Eysenbach *et al.* (2022) and Ghugare *et al.* (2022) considered jointly optimizing a global lower bound on policy performance with respect to both the model and policy parameters. Chow *et al.* (2020) proposed an EM algorithm to jointly improve the model and the policy with respect to a variational lower bound. One fundamental limitation of these approaches is that they do not address the suboptimalities introduced by the fact that models have limited representational capacity. In environments with complex dynamics that the model class is ill-suited to represent, a lower bound on policy performance may differ significantly from the true objective we wish to optimize (*i.e.*, $\mathcal{L}$ will be a loose bound for the true objective $J$), resulting in a poorly aligned policy-learning objective and suboptimal policies. Our approach builds on these ideas, but takes a fundamentally different approach: rather than using the model to approximate a single optimal policy, we maintain a set of policies that may be optimal, which is then refined by model-free RL, thereby avoiding over-reliance on potentially inaccurate models.

# 6 LIMITATIONS

Our approach has a few limitations that are worth noting. First, our approach does not incorporate intelligent exploration, and simply assume that the best policy at any given iteration is the ideal policy to collect new data, be it model-based or model-free. This assumption is potentially disadvantageous in environments that require extensive exploration, where short-term reward should be sacrificed for the purposes of information gain. This limitation could potentially be circumvented with a slight modification to the bound in equation 3 to include an exploration bonus corresponding to an approximation of the amount of information gained by executing a particular policy, similar to that used in Houthooft *et al.* (2016).

Another important limitaiton is that because Unified RL maintains two separate (model-based and model-free policies, but only collects data from one in a given episode, at least one of the two policies will be performing some amount of off-policy learning. This restricts our choice of model-free RL algorithm to off-policy algorithms, such as SAC or Q-learning. Even though SAC is in principle an off-policy algorithm, we found standard SAC to perform poorly when learning off-policy, requiring modifications to the Q learning process (Sec. 3.2) (Ball *et al.*, 2023). This limitation could potentialy be avoided by modifying the Unified RL algorithm to maintain one policy, that is updated with model-free RL, but constrained to lie within the equivalent policy set. This could be accomplished by incorporating a constraint into the model-free policy updates, similar to a trust region as used in PPO (Schulman *et al.*, 2017).

# 7 DISCUSSION AND FUTURE WORK

In this work, we propose *equivalent policy sets* (EPS), which we define as the set of policies that are not provably Bayes-suboptimal, according to bounds on policy performance constructed using a model. The EPS provides a valuable tool for quantifying how inaccuracies in the model translate into uncertainty in their estimate of the optimal policy. Using this tool, agents can better understand

in what situations models are useful, and when models should be abandoned in favor of model-free learning updates. Based on this concept, we proposed *Unified RL*, a novel RL algorithm that combines the relative strengths of model-based and model-free RL. Unified RL can be thought of as a model-free RL algorithm, where the enacted policy is constrained to lie within the EPS. Unified RL retains the data-efficiency benefits of model-based approaches by leveraging models to rule out provably suboptimal policies. However, by maintaining a set of candidate policies that *may* be optimal according to the model, which is then refined using MFRL, Unified RL avoids over-reliance on models and leverages the asymptotic performance benefits of MFRL. We show empirically on a wide range of challenging continuous control RL benchmarks that Unified RL successfully combines the strengths of both MBRL and MFRL, often exceeding the performance of either algorithm alone. We also find that Unified RL outperforms a number of state-of-the-art model-based and model-free prior approaches. Finally, we show that Unified RL learns effective policies in situations where either model-based or model-free RL alone fail.

In this work, we explore one strategy for combining MFRL and MBRL using the EPS, wherein the agent chooses between a model-based and model-free policy at each iteration. However, other alternative approaches exist, which we plan to explore in future work. One alternative that more tightly integrates the model-based and model-free components would be to incorporate the EPS constraint directly into the model-free learning updates. We also plan to explore using latent dynamics models, similar to those used in Dreamer, for Unified RL, as they have been shown to scale well to high-dimensional observation spaces and complex dynamics Hafner *et al.* (2019; 2020); Lin *et al.* (2023).

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

## A  APPENDIX

### A.1  DERIVATION OF LOWER BOUND IN EQUATION 3

The difference in Bayesian return between policies $\pi'$ and $\pi$ is given by

$$J(\pi'|D) - J(\pi|D) = \int_{\mathcal{W}} \frac{p(D|w)p(w)}{p(D)} \left( J(\pi'|w) - J(\pi|w) \right) dw. \tag{7}$$

By introducing an approximate posterior $q(w; \theta)$, we can write the above expression as an expectation over $q$,

$$J(\pi'|D) - J(\pi|D) = \mathbb{E}_q \left[ \frac{p(D|w)p(w)}{q(w;\theta)p(D)} \left( J(\pi'|w) - J(\pi|w) \right) \right]. \tag{8}$$

Let $\tilde{f}$ be a concave, monotonically increasing function. Taking $\tilde{f}$ of both sides and applying Jensen's inequality, we arrive at arrive at a lower bound on $\tilde{f}(J(\pi'|D) - J(\pi|D))$,

$$\tilde{f}(J(\pi'|D) - J(\pi|D)) = \tilde{f} \left( \mathbb{E}_q \left[ \frac{p(D|w)p(w)}{q(w;\theta)p(D)} \left( J(\pi'|w) - J(\pi|w) \right) \right] \right) \tag{9}$$

$$\geq \mathbb{E}_q \left[ \tilde{f} \left( \frac{p(D|w)p(w)}{q(w;\theta)p(D)} \left( J(\pi'|w) - J(\pi|w) \right) \right) \right]. \tag{10}$$

Finally, to arrive at $\mathcal{L}$, we define a new concave monotonically increasing function $f(x) = \tilde{f}(p(D)x)$ and substitute this into the above expression to eliminate the constant $p(D)$ term,

$$\tilde{f}(J(\pi'|D) - J(\pi|D)) \geq \mathbb{E}_q \left[ f \left( \frac{p(D|w)p(w)}{q(w;\theta)} \left( J(\pi'|w) - J(\pi|w) \right) \right) \right] \tag{11}$$

$$= \mathcal{L}(\pi, \pi', \theta, D). \tag{12}$$

To prove that $\pi'$ achieves higher Bayesian return than $\pi$, it is sufficient to show that $\mathcal{L}(\pi, \pi', \theta, D) > \tilde{f}(0) = f(0)$, thus the data likelihood term $p(D)$ is irrelevant in constructing the EPS.

## A.2 IMPLEMENTATION DETAILS

### A.2.1 MODEL ARCHITECTURE AND TRAINING

The dynamics model we used for all tasks consisted of a single linear input layer, followed by a single-directional, single-layer LSTM cell Hochreiter & Schmidhuber (1997), followed by two linear layers, and an output layer. The output layer consisted of four separate output heads, one each for state mean, reward mean, state standard deviation, and reward standard deviation. The standard deviation output heads used softplus activations to ensure their output was positive, while the mean layers did not an activation function. ReLU activations were used for all other layers other than the LSTM cell. State means were represented as learned deltas from previous states. That is, the state mean output predicts the mean in the *difference* between the current and last state. All inputs (states and actions) and outputs (state deltas and rewards) of the dynamics model were normalized before each period of model training to be of mean zero and unit variance.

Before each layer other than the initial input layer, including each internal layer within the LSTM cell, a binary dropout mask Srivastava *et al.* (2014) was applied, which was used by Gal & Ghahramani (2016a) and Gal & Ghahramani (2016b) to represent uncertainty in neural network parameters. Crucially, both in training and when sampling rollouts, the dropout mask is held fixed across all timesteps in a trajectory Gal & Ghahramani (2016b), while different dropout masks are sampled across trajectories. The dynamics model was trained with the following loss computed on a batch of trajectories sampled from the data buffer:

$$\mathcal{L}_{\text{model}} = \frac{1}{B} \sum_{i=1}^{B} \sum_{t=0}^{T} \left( \log p(s_{t+1}|s_t, a_t, w_i) + \log p(r_t|s_t, a_t, w_i) \right) + \frac{\eta}{N} ||W||_2^2 \qquad (13)$$

where $B = 100$ is the batch size, $T$ is the episode length, $w_i$ is the dropout mask corresponding to the $i$th trajectory, $\eta$ is a factor that determines the length-scale of the prior Gal & Ghahramani (2016b), N is the number of trajectories in the training dataset, Mand $W$ is the set of all learnable parameters in the network.

### A.2.2 POLICY ARCHITECTURE

Both model-based RL and SAC use a Tanh-Gaussian MLP policy with three layers with Tanh activations between layers. Policies used in MBRL have 1024 units in their hidden layers, while policies used for SAC have 256. The policies have two output heads, one for mean and one for standard deviation. The mean output head uses no activation function, while the standard deviation head uses either a softplus activation to ensure that standard deviation is positive, or a sigmoid activation to force the standard deviation to be bounded. To force samples from the policy to fall within the specified action range of the environment, samples are passed through a tanh function.

### A.2.3 CRITIC ARCHITECTURE

The critic network used for SAC was a state-action value function, while the critic used for MBRL was a state value function. In either case, critics consisted of 3 layers with ReLU activations between layers, with 256 units in each hidden layer.

### A.2.4 LOWER BOUND ESTIMATION

As discussed in Sec. A.2.4, to check whether the model-free policy $\pi^{MF}$ is within the EPS, we need only check whether $\hat{J}(\pi^{MB}|w_i) - \hat{J}(\pi^{MF}|w_i) > 0, \quad \forall i = 1, ..., K$, where $\hat{J}(\pi|w_i)$ is a Monte-Carlo estimate of $J(\pi|w_i)$, the expected return for policy $\pi$ given dynamics model parameters $w_i$. In the dropout formulation of BNNs, sampling $w_i$ corresponds to sampling a dropout mask, so we use $w_i$ to denote a particular dropout mask. Therefore, to compute $\hat{J}(\pi|w_i)$, we sample one dropout mask, and sample $M$ state-action-reward trajectories from our dynamics model and policy, from timesteps $t = 0$ to $T$ using that dropout mask, and average the return across those trajectories:

$$\hat{J}(\pi|w_i) = \frac{1}{M} \sum_{t=0}^{T} r_t, \qquad (14)$$

for $a_t \sim \pi(a_t|s_t)$, $r_t \sim p(r_t|s_t, a_t, w_i)$, and $s_{t+1} \sim p(s_{t+1}|s_t, a_t, w_i)$. Note that the dropout mask $w_i$ is held constant across timesteps.

### A.2.5 HYPERPARAMETERS

Table 2 contains the hyperparameters used for Unified RL for each task.

- $K$=Number of dropout masks sampled when computing $\hat{\mathcal{L}}$ (equation 6)

- $M$=Number of trajectories sampled when computing $\hat{\mathcal{L}}$ (equation 6)

- Policy training: whether the model-based policy is trained using full-trajectory policy training or Dreamer-style policy training, as described in Sec. 3.

- $\sigma_{max}$: In some cases, we found it useful to bound the maximum value that the policy standard deviation could take, by placing a sigmoid activation on the standard deviation output of the policy and multiplying by a constant. We refer to this upper bound as $\sigma_{max}$

- $\alpha_{MB}$: entropy bonus used for the model-based policy training

- $\alpha_{MF}$: entropy bonus used for SAC

- Automatic Entropy Tuning (MB policy): whether automatic entropy tuning is used for the model-based policy (makes $\alpha_{MB}$ irrelevant)

- Automatic Entropy Tuning (MF policy): whether automatic entropy tuning is used for the SAC policy (makes $\alpha_{MF}$ irrelevant)

- T: episode length

We additionally found it necessary to provide SAC with enough on-policy data by enforcing that at least one out of every 10 episodes was run using the MFRL policy.

Table 2: Hyperparameters used in Unified RL

| Environment | $K$ | $M$ | Policy Training | $\sigma_{max}$ | $\alpha_{MB}$ | $\alpha_{MF}$ | Automatic Entropy Tuning (MB policy) | Automatic Entropy Tuning (MF policy) | $T$ | $\eta$ |
|---|---|---|---|---|---|---|---|---|---|---|
| Ant | 50 | 5 | Dreamer | None | 0.2 | - | False | True | 100 | 200 |
| Hopper | 50 | 100 | Full trajectory | None | 0.2 | 0.2 | False | False | 200 | 100 |
| Walker | 50 | 100 | Full trajectory | 0.1 | 0.2 | 0.2 | False | False | 100 | 100 |
| Half Cheetah | 50 | 5 | Dreamer | None | 0.1 | - | False | True | 100 | 100 |
| Cartpole | 50 | default | Dreamer | None | 0.2 | - | False | True | 200 | 100 |
| DClaw-TurnFixed | 50 | 10 | Full Trajectory | None | 0.2 | 0.2 | False | False | 40 | 200 |

