6:     **if** $\hat{\mathcal{L}} > -\infty$ **then**
7:         $\pi = \pi^{MB}$
8:     **else**
9:         $\pi = \pi^{MF}$
10:     **end if**
11:     **for** time step t=0,...,T **do**
12:         $a_t \sim \pi(a_t|s_t)$
13:         $s_{t+1}, r_t = \text{env.step}(a_t)$
14:         $D \leftarrow D \cup \{s_t, a_t, r_t, s_{t+1}\}$
15:     **end for**
16: **end for**

---

We take a simple approach to combining model-based and model-free RL using the EPS, and leave more complex variants to future work. Before each episode, an MBRL and an off-policy MFRL algorithm use the available data $D$ to compute what we refer to as the *model-based policy* $\pi^{MB}$ and the *model-free policy* $\pi^{MF}$, respectively. Subsequently, the agent checks whether the model-free

policy is within the EPS; that is, it checks whether or not a lower bound can be constructed using the model that proves that the model-based policy achieves higher Bayesian return than the model-free policy. If the model-free policy is within the EPS, the agent executes it in the real environment to collect one episode of new data. If not, the agent instead executes the model-based policy, which is guaranteed to be within the EPS. The new data are then added to the shared data buffer, and the entire process repeats. Note that this approach does not require the EPS to be represented explicitly. Instead, the EPS is maintained *implicitly* in the sense that the lower bound in equation 2 provides a condition that allows one to check whether a given policy is within the EPS. We describe the individual components of our approach in more detail below, with additional details in Sec. A.2 of the Appendix.

**Model-Based RL** The MBRL component of our algorithm proceeds in two distinct steps: model training and policy training. During the model training step, we estimate the posterior parameters $\theta$ by fitting a Bayesian LSTM dynamics model to our environmental data $D$, by maximizing an evidence lower bound on data log likelihood (Kingma *et al.*, 2015; Gal & Ghahramani, 2016b;a),

$$\mathcal{L}_{\text{model}}(\theta, D) = \mathbb{E}_{w \sim q(w;\theta)} \left[ \sum_{i=1}^{|D|} \sum_{t=1}^{T} \log p(s_{t+1}^{(i)}, r_t^{(i)} | s_{\leq t}^{(i)}, a_{\leq t}^{(i)}, w) \right] - D_{KL}\big(q(w;\theta) || p(w)\big) . \quad (5)$$

Specifically, we use the *binary dropout* formulation of Bayesian LSTMs (Gal & Ghahramani, 2016a)~~to represent the dynamics model $p(s_{t+1}^{(i)}, r_t^{(i)} | s_{\leq t}^{(i)}, a_{\leq t}, w)$~~, wherein sampling a weight from the posterior $w \sim q(w;\theta)$ is accomplished by sampling a binary dropout mask from a fixed Bernoulli distribution (Gal & Ghahramani, 2016b). In this formulation, the prior $p(w)$ is approximately a Normal distribution, while the posterior is a Bernoulli Gal *et al.* (2016b). Our dynamics model ~~outputs the parameters for $p(s_{t+1}^{(i)}, r_t^{(i)} | s_{\leq t}^{(i)}, a_{\leq t}, w)$~~ is a Gaussian distribution over next state $s_{t+1}$ and reward $r_t$ with a diagonal covariance matrix, given the states $s_{\leq t}$ and actions $a_{\leq t}$ at all previous timesteps. The choice to represent state transition dynamics as Gaussians with diagonal covariance matrices is similar to past work ~~(Chua *et al.*, 2018; Gal *et al.*, 2016a; Gamboa Higuera *et al.*, 2018; Eysenbach *et al.*, 2022; Chow *et al.*, 2020)~~ (Gal *et al.*, 2016a; Chua *et al.*, 2018; Gamboa Higuera *et al.*, 2018; Chow *et al.*, 2020; Eysenbach *et al.*, 2022; Freed *et al.*, 202 , with the primary difference being that our dynamics model is recurrent. Specifically, we use an LSMT dynamics model, as we found this to yield more stable gradient-based policy optimization compared to a simple feed-forward MLP. ~~In the dropout formulation of Bayesian neural networks (BNNs), sampling a weight from the posterior $w \sim q(w;\theta)$ is accomplished by sampling a dropout mask from a fixed distribution (Gal & Ghahramani, 2016b).~~

$p(s^{(i)}_t + 1, r^{(i)}_t | s^{(i)}_{\leq} t, a_{\leq} t, w)$ . Our dynamics model outputs the parameters for