# OpenReview forum: "Unifying Model-Based and Model-Free Reinforcement Learning with Equivalent Policy Sets"
_ICLR.cc/2024/Conference — Submitted to ICLR 2024_

### Official Review · Reviewer_9w9H · 2023-10-28

**Soundness:** 3 good
**Presentation:** 3 good
**Contribution:** 3 good
**Rating:** 6
**Confidence:** 4

**Summary:**

The paper deals with the combination of model-free and model-based approaches in online reinforcement learning. A combination of both approaches is presented and tested on the basis of several benchmarks.

**Strengths:**

* The idea is original.
* The results are promising.

**Weaknesses:**

* The limitations of the approach remain unclear.

**Questions:**

1. Are stochastic MDPs among the benchmarks used?
2. How often were the experiments repeated in each case?
3. How were the uncertainties in Table 1 calculated?

Further comments:
* If "i.e." and "e.g." are written in italics, then consequently "et al." should also be in italics.
* Some of the uncertainties in Table 1 are given with too many digits. There should be one or two digits and not four as in "111.4". So actually "-176.9 ± 111.4" -> "(-18 ± 11) * 10" or, because this looks a bit messy in the table format and the 111.4 is the only uncertainty with four digits, "-176.9 ± 111.4" -> "-177 ± 111".
* Based on the statement "The characteristic feature of these approaches is an explicit representation of uncertainty in their estimate of the environmental dynamics. Gal et al. (2016) and Gamboa Higuera et al. (2018) are most similar to our approach" I would like to refer the authors to [1] and ask them to check how the similarity to [1] is.
* References contain some unintended lowercase: bayesian, pilco, rl

[1] S. Depeweg et al, Learning and policy search in stochastic dynamical systems with Bayesian neural networks, ICLR 2017.

---

### Official Review · Reviewer_Gnaz · 2023-11-01

**Soundness:** 2 fair
**Presentation:** 2 fair
**Contribution:** 2 fair
**Rating:** 5
**Confidence:** 4

**Summary:**

This paper proposes a novel reinforcement learning (RL) algorithm, named Unified RL (URL), which introduces a way to switch between policies learned via model-free and model-based methods. To achieve this goal, the authors propose equivalent policy sets (EPS), which is the set of all policies for which there does not exist a provably better policy in terms of Bayesian return (expected return over the learned dynamics model parameter).
In practical terms, URL works by learning a policy with a model-free algorithm, and a second policy using model-generated transitions. Then, the returns of both policies are evaluated via Monte Carlo estimation using the learned dynamics model with different dropout masks. The model-based policy is selected only if its return is greater than the return of the model-free policy when evaluated in all models/dropout masks. URL is evaluated in robotics tasks and compared with state-of-the-art model-free and model-based RL algorithms.

**Strengths:**

* The idea of exploiting the benefits of model-based and model-free algorithms in combination is a very relevant topic in the RL field.
* Based on the experimental results (e.g. Figure 3), the proposed method is able to switch between model-based and model-free policies when more appropriate.

**Weaknesses:**

* It is not clear how the method handles function approximation errors in the model. For instance, if the learned model generates overestimated rewards, it could be possible that the model-based policy learns to outperform the model-free policy when evaluated in the model, even though the model-free policy would perform better in the real environment. This could lead to the model-based policy being incorrectly selected.

* The practical URL (Algorithm 1) only superficially uses the theoretical idea of EPS (Eq. 4). While EPS is the definition of a set of policies for which we can not identify provably better policies using the model, URL only considers a single model-free policy. Moreover, as mentioned above, it is not clear how can we rely on the Bayesian returns when the model is inaccurate.

* The method introduces significant computational overhead, as it requires running two algorithms (one model-free and one model-based) simultaneously. Furthermore, the introduced overhead does not result in significant performance improvements (see Figure 2).

**Questions:**

Below, I have a few questions and constructive feedback to the authors:

The EPS is defined as the *smallest* possible set of policies that are not provably Bayes-suboptimal. Why is it the smallest set? Also related to this question, what is the domain of $\pi’$ in the maximization in Eq. (4)? Is it the set of all possibly existing policies?

“In environments where either model-based or model-free RL strictly dominates, Unified RL matches or outperforms the better algorithm.”
This is not true from the Walker or Cartpole results, as the PPO is the best-performing algorithm in these domains.

Why do the results of the ALM algorithm have such high variance and strange learning curves? Are these results comparable with the results of the work that introduced ALM?

In Figure 3 (Left), why does URL perform slightly better than SAC? If the MBRL policy is unable to solve the task because of the distractors, it would be expected that the MFRL is always selected, thus they should have the same performance.

The authors claim that “rather than using the model to approximate a single optimal policy, we maintain a set of policies that may be optimal, which is then refined by model-free RL, thereby avoiding over-reliance on potentially inaccurate models.” It is not clear to me how this is true. The algorithm only maintains one model-free and one model-based policy. How are a set of policies refined by model-free RL?

Minor:
 * In the caption of Figure 3, it should be (Left) and (Right) instead of (Top) and (Bottom)
 * “challenging continues control tasks” -> continuous

---

### Official Review · Reviewer_1YmJ · 2023-11-01

**Soundness:** 2 fair
**Presentation:** 3 good
**Contribution:** 2 fair
**Rating:** 5
**Confidence:** 4

**Summary:**

# Summary
This work first introduces the concept of EPS (equivalent policy sets), which is used to compare whether a given candidate policy is provably suboptimal compared to a reference policy. Then, an algorithm is proposed that approximates this condition for two policies as input: (i) a candidate policy learned in a model-free soft-actor-critic (SAC) algorithm, and (ii) a reference policy learned in a model-based RL algorithm (`MBRL` baseline). If the condition holds, then the candidate policy is provably sub-optimal to the reference policy and the algorithm decides to use the reference policy for interacting with the environment. Otherwise, the candidate policy is utilized for this purpose.

Empirical analysis is performed to compare the proposed strategy to prior model-based and model-free methods as well as ablations of the proposed method in the form of an MBRL and SAC baseline which constitute the model-based and model-free algorithms used in the proposed method respectively. The analysis is focused on highlighting cases where the algorithm is able to keep the best performance of its individual components (MBRL and SAC), especially in cases where either component alone is expected to fail.

**Strengths:**

# Strengths
TL; DR: Novel idea, sound theory.

1. The proposed tool -- Equivalent Policy Sets, is a strong contribution in isolation (keeping aside Unified RL). Equation 6, which approximates the asymmetric sub-optimality check of two policies, seems like it can be widely used in several RL algorithms that maintain multiple policies, beyond what is presented in this paper. For example, this could be used to compare policies within an ensemble of model-free policies given a reference model-based policies. The theory behind it is sound and the conditions for equality and the intricate consequences of the approximation of this sub-optimality condition are well explained in this work.

1. Unified RL, that is stated to be the simplest realization of EPS in practice, is a good enough (i.e. minimum sufficient) way of testing the effectiveness of EPS -- it uses just one MBRL policy and MF policy. The practical implication of using the approximation of (Eqn 6) -- that the ML policy will be selected more often in Algorithm 1 -- is acknowledged.

1. The design of Unified RL naturally leads to a strength -- the objective mismatch problem of model-based RL is avoided.

1. The ablation experiment for robustness to both model misalignment and excessively high model-free policy entropy clearly show that Unified RL is performing exactly as intended in it’s design -- it maintains at least the better performance of its components i.e. performance(Unified RL) >= max(performance(MFRL), performance(MBRL).

**Weaknesses:**

# Weaknesses
TL; DR: Empirical evidence not convincing.

1. The empirical evidence demonstrates that performance(Unified RL) >= max(performance(MFRL), performance(MBRL). However, the performance of Unified RL is shown to be just slightly higher than the max of the two components -- being higher in just the Hopper environment and slightly higher in the Walker environment (out of the 6 environments presented). Otherwise, it seems that equality holds. This is problematic as the choices of environments in this paper are not representative of environments where it is absolutely necessary to shift between MF and MBRL multiple times during training. In most environments, it seems that either MF or MBRL is a clear winner. Since most of RL literature (including this work) “trains on the test set”  i.e.: hyperparams are tuned on each environment and then reward curves are shown on the same environments -- one may simply choose argmax(performance(MFRL), performance(MBRL) and not have that much worse performance than Unified RL on each environment. This empirical evidence needs to convince us of the importance of using Unified RL vs argmax(performance(MFRL), performance(MBRL).

1. The paper uses the phrase -- “maintaining a set of candidate policies” multiple times. However, the proposed algorithm (Unified RL) never maintains such a set, as it would be intractable -- instead, it just compares two policies (MF and MBRL policies). This is a misleading phrase as it inflates the capabilities of Unified RL.

1. The definition of EPS also seems to either be wrong or have a crucial typo -- shouldn’t it be defined as the *largest* possible set of policies that are not provable Bayes-optimal (…) instead of *smallest* possible set of all policies?

1. ALM baseline HalfCheetah and Ant results don’t seem to match the cited ALM paper. This seems like a serious issue that may invalidate some of the conclusions drawn.

**Questions:**

# Questions and Suggestions

1. Correct me if I am wrong, but given that L-hat(pi-MF, pi-MB, …) > minus-infinity, shouldn’t pi-MB be the selected policy in the if-condition of Algorithm 1 as there is a positive lower bound to the performance difference of pi-MB - pi-MF?

1. Can we see plots with number of steps instead of number of episodes on the X-axis for comparison with other works?

1. What version of halfcheetah and other OpenAI gym environments is used? Is it HalfCheetah-v2?

1. I strongly recommend expanding the types and number of environments for empirical evaluation.

1. Does the SAC baseline use the two modifications mentioned in the SAC used for the proposed method (i.e. layer normalization in Q networks and omission of entropy term for Q-net loss)?

1. I think the future work mentioned in Section 6 seems promising!

---

### Official Review · Reviewer_jTrP · 2023-11-02

**Soundness:** 3 good
**Presentation:** 3 good
**Contribution:** 2 fair
**Rating:** 3
**Confidence:** 3

**Summary:**

The paper studies the combination of model-based RL and model-free RL. The authors propose an approach that makes use of the concept of equivalent policy set (EPS) that, based on a Bayesian formulation, represents the set of policies that are not provably Bayes-suboptimal according to the current data and prior. The algorithmic contribution, Unified RL, switches between model-free and model-based RL according to the value of a variational lower bound which is evaluated from the prior and the model-based and model-free policies. The paper provides experimental validation on the set of Mujoco environments and a validation for testing the robustness to misalignment.

**Strengths:**

- The idea of combining model-free and model-based RL is surely relevant in the RL community.
- The paper introduces the novel concept of equivalent policy set which has a nice interpretation from a Bayesian perspective.
- The paper is well written and the contributions are clearly outlined.

**Weaknesses:**

- [Choice of the function $f$] The expression that is used to evaluate when to switch between model-based and model-free policies is based on a function $f$ (eq. 3) which is not further specified (apart from the fact of being concave and increasing). It is not clear to me why you are allowed to choose $f$ arbitrarily (given that it is concave and increasing). Can the authors elaborate? Furthermore, in the experimental section, which $f$ is used?

- [Choice of $p$ and $q$] These elements represent the prior and the approximate posterior that is used in the algorithm to evaluate the loss function for selecting when to switch. How are they selected? $p$ should be a property of the formalization of the problem in the Bayesian context. In the experimental part, how is it selected? Furthermore, $q$ is the approximate posterior and, I believe, its choice greatly influences the performance of the algorithm. In which class of probability distribution is picked?

**Minor Issues**
- The plots do not report the number of runs and the meaning of the shaded areas (std, confidence intervals)
- Multiple citations should be in chronological order.

**Questions:**

Please refer to [Weaknesses].

---

### Author Response · Authors · 2023-11-23
**Response to Reviewer jTrP**

We thank the reviewer for their thoughtful and constructive feedback.  In response to the weaknesses raised:

[Choice of function f] An arbitrary concave f can be chosen because our bound is derived using Jensen’s inequality[1], which applies to any concave f.  It is true that this choice of f may have a significant impact on the performance of the algorithm.  However, this is also a feature of all variational inference-based approaches, with several papers proposing new variants on VI that use different f’s when deriving their evidence lower bounds [2, 3, 4, 5].  Because choice of f is a rich area of research in its own right, we left evaluating the effects of f to future work.  We chose to use log for f, as mentioned in the Lower Bound Estimation subsection of Sec. 3.2, because log is the most well-studied choice of f for VI [6,7].  We have clarified this choice of f in the manuscript.

[Choice of p and q] The distribution family from which p and q are chosen is implicitly decided by our choice of Bayesian neural network formulation.  As mentioned in Sec. 3.2, we chose to use binary dropout Bayesian neural networks, proposed by [8], due to their simplicity, computational efficiency, and the fact that they have been shown to perform well in an MBRL context [9, 10].  We also chose p and q to be as generic as possible, so that we encode minimal prior knowledge into our models.  For the type of BNN that we used, p approximately corresponds to a normal distribution, while q is a Bernoulli distribution [11].  In terms of how a _particular_ p and q were chosen for a given problem, p has one free parameter (the prior length-scale) which was treated as a hyperparameter.  To minimize fine-tuning for particular environments, all environments use one of only two different length scales (see eta column, Table 2 in the appendix).  q was trained in a standard Bayesian supervised learning fashion on the available data using variational inference (see Model-Based RL subsection in Sec. 3.2).  We have clarified these points in the manuscript, and have included choice of prior in our formalization of the problem in the Bayesian context.

In response to the minor issues raised:
- Thank you for pointing out this oversight on our part.  All results are computed by averaging over 5 distinct random seeds.  Shaded areas on plots are 95% confidence intervals.  We have clarified this in the text.
- We have corrected the order of citations in the manuscript.

[1] Weisstein, Eric W. "Jensen's Inequality." From MathWorld--A Wolfram Web Resource. https://mathworld.wolfram.com/JensensInequality.html

[2] Li, Yingzhen, and Richard E. Turner. "Rényi divergence variational inference." Advances in neural information processing systems 29 (2016).

[3] Wan, Neng, Dapeng Li, and Naira Hovakimyan. "F-divergence variational inference." Advances in neural information processing systems 33 (2020): 17370-17379.

[4] Adji Bousso Dieng, Dustin Tran, Rajesh Ranganath, John Paisley, and David Blei. Variational inference via χ upper bound minimization. Advances in Neural Information Processing Systems, 30, 2017.

[5] Liqun Chen, Chenyang Tao, Ruiyi Zhang, Ricardo Henao, and Lawrence Carin Duke. Variational inference and model selection with generalized evidence bounds. In International conference on machine learning, pp. 893–902. PMLR, 2018.

[6] Kingma, Diederik P., and Max Welling. "Auto-encoding variational bayes." arXiv preprint arXiv:1312.6114 (2013).

[7] Blei, David M., Alp Kucukelbir, and Jon D. McAuliffe. "Variational inference: A review for statisticians." Journal of the American statistical Association 112.518 (2017): 859-877.

[8] Li, Yingzhen, and Yarin Gal. "Dropout inference in bayesian neural networks with alpha-divergences." International conference on machine learning. PMLR, 2017.

[9] Gal, Yarin, Rowan McAllister, and Carl Edward Rasmussen. "Improving PILCO with Bayesian neural network dynamics models." Data-efficient machine learning workshop, ICML. Vol. 4. No. 34. 2016.

[10] Depeweg, Stefan, et al. "Learning and policy search in stochastic dynamical systems with bayesian neural networks." arXiv preprint arXiv:1605.07127 (2016).

[11] Gal, Yarin. "Uncertainty in deep learning." (2016): 3.

---

### Author Response · Authors · 2023-11-23
**Response to Reviewer 1YmJ**

We thank the reviewer for their in-depth and constructive feedback.  In response to the weaknesses raised:

1. We would like to push back on the statement that Unified RL only achieves better performance in Hopper and Walker environments. We performed a t-test to determine whether Unified RL significantly improved over both SAC and MBRL in terms of cumulative reward throughout the training period, and found that the improvement over SAC was significant in 4 out of the 6 environments (all but Cartpole and DCLawTurnFixed), and that the improvement over MBRL was significant in 5 out of the 6 environments (all but DCLawTurnFixed) .  Additionally, Unified RL was the most consistently high-performing algorithm out of all the algorithms we tested.  To clarify this point, we have added the average rank that each algorithm achieved, which we also include below:

Average cumulative reward rank (lower is better) across all environments:

Unified RL (ours): **2.33**

SAC: 3.5

MBRL: 4.5

ALM: 7.33

DDPG: 6.83

SVG: 4.83

TD3: 5.83

HL: 4.16

PPO: 5.66

Unified RL achieves the highest average rank across all environments, indicating it is the most consistently high-performing algorithm that we tested.

2. It is true that we do not _explicitly_ represent the set of policies comprising the EPS, as you are correct that this would be intractable.  However, it is not necessary to explicitly represent the EPS given that our goal is to constrain the policy of Unified RL to be within the EPS.  For this, we require only a condition that allows us to check whether a policy is within the EPS, which we accomplish using the bound in eq. 6.  We would therefore argue that we do  _implicitly_ consider the full set of policies that are not provably suboptimal.  We have modified the manuscript to reflect this point.
3. Thank you for pointing out this inaccurate wording.  It would indeed be more accurate to say that the EPS is the set of _all_ policies that are not provably Bayes-suboptimal.  Our intent was to convey the fact that the EPS is the smallest set of policies pi for with a new policy, pi’ provably achieves higher Bayesian return (hence the maximization over pi’ in eq. (4)).  This is because we wish to exclude as many suboptimal policies from the EPS as possible.  However, we acknowledge that our wording in the paper is unclear and we have corrected this issue.
4. The results we report for ALM were obtained by running the authors’ released code with the recommended hyperparameters.  It is possible that the deterioration in performance of ALM resulted from the fact that our experiments used shorter episode lengths (100-200 timesteps vs. 1000 timesteps).  We have clarified this in the manuscript.

Responses to questions:
1. You are correct, this was a typo.  We have corrected this in the manuscript.
2. Yes, we have changed the x axis on the plot from episodes to timesteps.
3. Version 2
5. Yes, for the most apples-to-apples comparison we ran SAC with the same two modifications that were used in Unified RL (i.e., layer normalization and removal of entropy backups), though we found these modifications to not have a major impact on the algorithm’s on-policy performance.
6. Thank you, we appreciate that feedback!

---

### Author Response · Authors · 2023-11-23
**Response to Reviewer Gnaz**

We thank the reviewer for their thoughtful and constructive feedback.  In response to the weaknesses raised:
- One of the primary contributions of our approach is a principled method for dealing with function approximation errors in the model.  We fully agree that modeling error may cause the model to overestimate the return of the model-based policy.  However, because we use a _lower bound_ on the improvement of the MB policy over the MF policy, Unified RL _underestimates_ this improvement during the policy selection step.  The lower bound in eq. 3 is an _exact_ lower bound on improvement in Bayesian return of pi’ over pi.  Because function approximation errors cause this lower bound to be loose (and therefore lower), these errors bias the agent _away_ from the MB policy.  The way this practically manifests in the algorithm is as follows: when function approximation error exists (e.g., when our model class is misspecified), the approximate posterior assigns probability mass to the set of candidate dynamics models that best explain the data while not being too improbable a priori.  We should expect the approximate posterior to assign some mass to the dynamics models that underestimate as well as overestimate the return of the model-based policy.  When it comes time for our agent to select a policy using a Monte-Carlo estimate of the lower bound on policy improvement, due to the specific form of the bound we use, the agent will only select the MB policy if it yields higher predicted reward according to _all_ dynamics models sampled from the posterior.  Therefore, even if the approximate posterior overestimates the improvement of the MB policy over the MF policy _on average_, the agent will select the MF policy if at least one of the sampled dynamics models predicts that the MF policy will outperform the MB policy.
- It is true that we do not _explicitly_ represent the set of policies comprising the EPS, as this would be intractable.  However, it is not necessary to explicitly represent the EPS given that our goal is to constrain the policy of unified RL to lie within the EPS.  For this, all we need is a condition that allows us to check whether a policy is within the EPS, which we accomplish using the bound in eq. 6.  We do therefore consider the entire set of policies that are not provably suboptimal, in the sense that model-free RL can choose any policy within this set.  It is true that our method of constraining the policy to lie within the EPS is simplistic; however, we thought it best to favor simplicity in our approach given that this is the first work on the topic.
- It is true that our method introduces additional computational overhead.  However, we argue that for most real-world systems, the data efficiency gains more than compensate for this increased computational overhead.  For example, in the Ant environment, to achieve 80% of its maximal reward, SAC requires 2480 episodes, while Unified RL requires only 1404 episodes.  SAC requires 0.008s of wall clock time per episode to update the policy, while time taken to collect one episode of data in the real environment is on the order of 1s (assuming our RL controller operates at 100Hz, which is the frequency used by the Mujoco simulator).  Under these assumptions, we can tolerate a 97-fold increase in computational burden and still require less wall clock time to reach 80% of maximal performance, because the time required to perform learning updates is practically negligible compared to the time required to collect data.

---

### Author Response · Authors · 2023-11-23
**Response to Reviewer Gnaz (Continued)**

To address your questions:
- Thank you for pointing out this inaccurate wording when we referred to the EPS as the smallest possible set of policies that are not provably suboptimal.  It would be more accurate to say either that the EPS is the set of all policies that are not provably Bayes-suboptimal using a particular model, or that the EPS is the smallest set of policies pi for which a new policy pi' is not guaranteed to improve over pi.  In this latter description, the word _smallest_ appears because we perform an optimization over pi’ to try to rule out as many policies pi as possible.
- When we said that “in environments where either model-based or model-free RL strictly dominates, Unified RL matches or outperforms the better algorithm,” we meant the better performing algorithm between the model-based and model-free components of Unified RL.  We have clarified this in the manuscript.
- Results for ALM were achieved by running the code released by the authors with the recommended hyperparameters for each environment.  Differences between results reported by the authors and our results could be due to the fact that we use shorter episode lengths (100-200 timesteps vs. 1000 timesteps), which might hinder the Q learning step of ALM.  We have clarified this in the manuscript.
- We do not know with certainty why Unified RL slightly outperforms SAC in the environment with distractors.  One possible explanation is that Unified RL is able to gain some small benefit from using the model, perhaps because it has been trained on data collected from a higher performing policy and may therefore be more task-aligned, while the model used by MBRL alone never trains on data from a high-performing policy.
- What we meant by “rather than using the model to approximate a single optimal policy, we maintain a set of policies that may be optimal, which is then refined by model-free RL, thereby avoiding over-reliance on potentially inaccurate models”, is that we do not simply use the policy that is optimal according to our model.  Instead, we _implicitly_ maintain the set of policies that are not provably suboptimal, and use the _model-free policy_ whenever it lies within this set.  Otherwise, we use a policy that is guaranteed to be within the equivalent policy set (the model-based policy).  Therefore, unified RL can be viewed as a model-free RL algorithm where the policy is constrained to lie within the equivalent policy set.  We have clarified this in the manuscript.

---

### Author Response · Authors · 2023-11-23
**Response to Reviewer 9w9H**

We thank the reviewer for their insightful feedback.  Indeed a discussion of limitations is important.  We have added a section on limitations to the manuscript where we discuss the following:
- Currently, our approach does not address exploration.  In both the theory and implementation, we essentially assume that the policy that achieves the highest return is also the policy that is best for collecting more data.  This is a common assumption in reinforcement learning; however, in certain cases it may be beneficial to execute a policy that achieves lower reward but provides more information about how to best to increase reward in the future.  To address this issue, we could introduce an information-gain bonus into eq. 3 that preferences the agent towards policies for which our model has a high degree of epistemic uncertainty.
- Unified RL has the drawback that it maintains two separate policies (one model-free and one model-based), meaning that at least one of these policies must perform some amount of off-policy learning.  This limits the set of model-free RL algorithms we can use to off-policy algorithms, such as SAC.  Furthermore, empirically SAC has been shown to perform poorly off-policy, necessitating modifications [2].  This could be addressed in our approach by instead maintaining only one policy, which we update by performing a constrained optimization of a model-free RL objective.  Here, the constraint would force the resulting policy to lie within the EPS.

To answer your questions:
1. We did not consider any stochastic MDPs.  Are there any that you suggest?
2. Experiments are each repeated 5 times with distinct random seeds.
3. Uncertainties in table 1 are 95% confidence intervals (1.96*standard_error)

We have also addressed your comments in the revised manuscript.  Thank you for pointing out this highly relevant related work [1].  We have added this reference to our manuscript.  Indeed the approach in [1] is very similar to the MBRL component of our algorithm, with the primary differences being that we use a slightly different Bayesian neural network formulation and use KL divergence as opposed to alpha divergence to fit our approximate posterior.

[2] Ball, Philip J., et al. "Efficient online reinforcement learning with offline data." arXiv preprint arXiv:2302.02948 (2023).

---

> ### Comment · Reviewer_9w9H · 2023-11-23
>
> Thank you for your answer. The explanations are helpful to better assess the paper.
> > Uncertainties in table 1 are 95% confidence intervals (1.96*standard_error)
>
> This is very good, and also important, that it is now explicitly stated in the caption---unfortunately there are repeatedly publications in the ML literature that, contrary to all standards, state the standard deviation where an uncertainty is required.
>
> Naming the limitations also significantly improved the paper in my opinion.
>
> > We did not consider any stochastic MDPs. Are there any that you suggest?
>
> For future work, it could be an idea to test the method on a stochastic MDP. There are many benchmarks with stochastic MDPs, e.g. those used in Depeweg et al. (2017): the just two-dimensional 2D wet chicken benchmark or the high-dimensional "industrial benchmark". Stochastic MDPs are also included in the benchmark suits RL Unplugged [1] and NeoRL [2]. I would also like to refer to [3] on the subject of the one-sided benchmark selection.
>
> Overall, I now see the paper more positively and am in favor of accepting it.
>
> [1] C. Gulcehre et al., RL Unplugged: A Suite of Benchmarks for Offline Reinforcement Learning, 2020\
> [2]  R.-J. Qin et al., NeoRL: A Near Real-World Benchmark for Offline Reinforcement Learning, 2022\
> [3] S. Mannor and A. Tamar, Towards Deployable RL -- What's Broken with RL Research and a Potential Fix, 2023

---

### Author Response · Authors · 2023-11-23
**Revised manuscript**

We have added a revised manuscript that incorporates reviewers' feedback.  We additionally included a diff file as supplementary material that highlights the differences between the original and revised manuscript.  We would once again like to thank the reviewers for their constructive feedback.

---

### Meta-Review · Area_Chair_k8Ru · 2023-12-15

**Metareview:**

This paper proposes the idea of equivalent policy sets to combine model-based RL and model-free RL. All the reviewers agree that the idea of EPS is novel.

While the authors clarified several concerns during the rebuttal, two of the reviewers questioned the significance of empirical claims in the paper. While the proposed method is compute-intensive two train two agents at a time, the performance gain is not very high when compared to max(MF, MB). The authors provided the ranking of the models. But I don't think ranking gives the complete picture. This is a weakness the authors must address to convince others of the usefulness of the proposed method. one suggestion is to find environments where both model-based and model-free methods struggle and show that your method can actually solve the task.

I recommend authors to improve their experiments and resubmit.

**Justification For Why Not Higher Score:**

Experiments are weak.

**Justification For Why Not Lower Score:**

N/A

---

### Decision · Program_Chairs · 2024-01-16

Reject